# Pharmaco-Analysis of Psychedelics—Philo-Fictions about New Materialism, Quantum Mechanics, Information Science, and the Philosophy of Immanence

Stefan Paulus 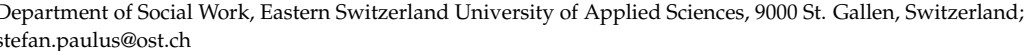

Department of Social Work, Eastern Switzerland University of Applied Sciences, 9000 St. Gallen, Switzerland; stefan.paulus@ost.ch

**Abstract:** Recent developments regarding the pharmacology of psychoactive substances are significant for treating depressions or opioid addictions. Current theories, hypotheses, and models of drug effects assume a cause–effect narrative, which is based on a stimulus/response mechanism. These narratives prioritize effects rather than conscious experiences. In this sense, drug experiences are quickly subsumed into common categories and codes of biological determinism. If subjective experiences are in the focus of the research, it quickly becomes a link to mystical, spiritual, or transcendental narratives. These classifications lead to epistemological doublets (Gadamer). In this article, psychedelic experiences of drug users are analyzed in the frame of the pharmaco-analysis by Deleuze/Guattari. These framed psychedelic experiences are interpreted by means of a non-philosophical approach through philo-fictions (Laruelle), i.e., contradictory assumptions and hyperspeculations. In this respect, the aim of this article is to bring philo-fictions in relation to psychedelic experiences and to discuss them with models of information science, quantum mechanics, new materialism, and the philosophy of immanence. The result will be an open synthesis, with the assumption of further reflections on the agency, immanence, and the wholeness of matter.

**Keywords:** philo-fiction; information science; quantum mechanics; new materialism; immanence

## 1. Introduction

In the context of my subject-scientific research activities with persons suffering from burnout depression and their treating physicians, psychiatrists, and social workers, I explore different possibilities of prevention, treatment, and aftercare of stress [1,2]. In interviews or workshops, the advantages and disadvantages of medication are also a frequent topic of discussion. In these settings, the possibilities of therapeutic use of psychedelic resp. entheogens [3] such as psylocibin, DMT or dissociatives such as ketamine were discussed at different points.

The evidence that psychedelic substances, such as psilocybin, DMT or LSD, are effective against depression, stress, and anxiety is highly significant that in the US, for example, Johns Hopkins University has opened a Center for Psychedelic and Consciousness Research, psychotherapists are being trained in the use of psychedelics at the Center for Psychedelic Therapies and Research in San Francisco, and the US Food and Drug Administration has granted the substance psilocybin a breakthrough therapy status as a therapeutic for depressive disorders. As a result, the development of drugs based on psychedelic substances is considered by the US Food and Drug Administration to be particularly urgent [4–7].

In general, DMT (Dimethyltryptamine) is a psychedelic of the tryptamine class. Parts of the structure of DMT occur within biomolecules like serotonin or melatonin. There are functional and structural analogs of psychedelic tryptamines such as psilocybin (4-PO-DMT), psilocin (4-HO-DMT), bufotenin (5-HO-DMT), 5-MeO-DMT (5-Methoxy-*N*,*N*-DMT), and *N*,*N*-DMT (*N*,*N*-DMT). *N*,*N*-DMT is a naturally occurring alkaloid, which can be found in plants, animals, and humans. Bufotenin and 5-MeO-DMT are found in a smaller number

of plant species and animals. Psilocybin and psilocin can be found in fungi. The difference between the substances can be found in the potency of the tryptamine and duration of the experience. 5-MeO-DMT is considered as one of the most potent hallucinogens and has been used as an entheogen in the US since Mayan culture. It is found, among other sources, in seeds of leguminous plants. Additionally, 5-MeO-DMT is found in the skin secretions of some toad species, such as Bufo Alvarius. Especially for psychiatric use, 5-MeO-DMT seems to have potential: "those who reported being diagnosed with psychiatric disorders, the majority reported improvements in symptoms following 5-MeO-DMT use, including improvements related to post-traumatic stress disorder (79%), depression (77%), anxiety (69%), and alcoholism (66%) or drug use disorder (60%)". [7] I had the opportunity to be present at a treatment of 5-MeO-DMT in a professional setting and had the chance to interview drug users (see Section 2). For this reason, unless otherwise described, the article highlights the research about 5-MeO-DMT, or refers to DMT in general as a psychedelic of the tryptamine class.

Currently, the long duration of the substances is still problematic for their use in psychotherapy. Depending on the dose, the effect of psilocybin or LSD can last 3–10 h. However, the duration of 5-MeO-DMT is much shorter (15–90 min) and seems to be more suitable for a broad use in psychotherapy. When 5-MeO-DMT is inhaled, the effect occurs within seconds. Reports of experiences describe encounters with God, enlightenment experiences, the feeling of being reborn, or the feeling of becoming one with the universe, as well as near-death experiences [8,9].

The new developments in terms of the pharmacology of psychoactive substances are curious and noteworthy, but no one and certainly not persons with depression, should be expected to have near-death experiences. This raises the question of why psychedelic-assisted psychotherapy seems promising. The spectrum of effects of psychedelic substances seems even more promising that universities and pharmaceutical companies are focusing their research on bringing appropriate drugs to the market, e.g., with shorter duration or the reduction in undesirable effects, such as hallucinations. Meanwhile, there is an ongoing narrative of a psychedelic renaissance in pharmacology [10]. At a time when every fourth European is at risk of suffering from stress-induced depression (burnout) at least one time in his or her working life [2,11], the efficacy of psychedelics also seems to fit the narrative of the post Fordist working society: If psychedelic drugs served in the rebellious counterculture of the 1960s to break out and to live a life beyond social conventions, psychedelic drugs are now intended for neuro-enhancement, performance enhancement, and precisely to be able to reintegrate into the world of work. Therefore, the new spirit of capitalism prepares an access to substances that serve to enable the subject to optimize itself [12].

Clinical and pharmacological studies explore less the subjective experiences or philosophical dimensions of psychedelic substances. Instead, the effects on the organism are deciphered to more specifically isolate mechanisms of action and control appropriate tolerances, dosages, and side effects. Psychedelic drugs have been proven to influence serotonin levels, as well as tricyclic antidepressants, such as Prozac. It is also still unclear whether fluctuations in serotonin levels are responsible for the antidepressant effect or even a change in the living and working situation [13]. The implication is that there is still a quest to answer the question of why psychedelic drugs are effective in combating depression.

Questions in pharmacological studies on whether psychedelic substances produce their full effects only in a specific cultural setting or if the function of the integration of the experience into the everyday life is important are less asked here. A cause-and-effect narrative of drug impact is attempted, based on a simple stimulus/response mechanism specified at an intentional operational level of a regulative interaction between anticipation and event production. In this way, pharmacologically regulated operations are supposed to refer to superordinate social goal constellations whose partial goals are these operations themselves [14], namely, the production of a functional organism or social body.

Furthermore, theories based on theological or metaphysical views assume that the self is no longer the processor of substance and the cause of experience, but that a spiritual entity transcends itself through substance [15]. Drug experiences here are quickly absorbed into or assigned to the common categories and codes of spirituality. The term entheogen was specifically invented for this purpose [3,15–18]. Even pharmacological studies and research projects investigate the thesis that mystical experiences have a long-term effect on well-being. But in a meta-analysis by McCulloch et al., the researchers assume self-critically that qualitative analyses are often limited by biased selection of quotes that fit the narrative of the researchers [19] (p. 14).

- Question

Philosophically, therefore, the question is how to analyze the phenomenon of a psychedelic drug experience in its multiple dimensions without resorting to epistemological duplications. This includes the action-theoretical question of whether the subject or the substance is the processor or cause of the experience, as well as the question of whether the praxeology of the drug experience itself does offer the possibility of extracting explanations of drug experiences that go beyond the current codes of the effects of the drug.

To methodologically approach these three questions: (a) question of experienced dimensions, (b) causation of the mental state, and (c) of the materiality of experience, there will be a recourse to the philosophical pharmaco-analysis of Gille Deleuze and Félix Guattari, as well as to the experimental or non-philosophical method of philo-fiction of François Laruelle.

- Methodological Approach

Deleuze and Guattari describe the pharmaco-analysis as an investigation of experiences of consciousness that split off from the molar perception [20] (p. 60f), i.e., experiences that are qualitatively different from experiences within the order of everyday experience, of being mired in a functional everyday life. These experiences lead to realms of the unconscious, the molecular perception. For Deleuze and Guattari, perception is seen as an activity shaped by socio-political influences, in which different things interlock, overlap, block, catalyze, interrupt, etc. To explain this, distinctions are made between molar and molecular lines or types of perception. The molar line strictly separates areas from each other, e.g., perception of work/leisure and family/profession, including the corresponding perception schemata as father, mother, worker, boss, etc. and the corresponding action schemata as father, mother, worker, boss. The molecular is not present in the self-conception of father, mother, worker, boss. The molecular can best be recognized in a perception or a body that is liberated from the molar. In Anti-Oedipus (1977) and A Thousand Plateaus (1987), Deleuze and Guattari developed the figure and the call to action of the Body without Organs (BwO). The BwO can be understood as a concept that implies a psychophysical experience of the self through a critique of societal modes of subjectification, of a de-subjectivation. Accordingly, in order to create a BwO, it is necessary to eliminate the meanings and regularities of society and to fill oneself with intensities. For Deleuze and Guattari, life is more intense the more it is molecular and inorganic. Dissolving the organism and becoming intense can also be understood in a figurative sense, not by killing the body but by developing multiplications, superimpositions of meanings and codes, and by going into the unknown, the undescribed, the unorganized, the formless of the desert, the high mountains or psychedelic experiences. Deleuze and Guattari suggest that this aggregation of intensity to smooth, reduce, and remove subjectivity takes place in ecstasy, experimentation, sound production, and movement. Deleuze and Guattari describe the BwO as a variation of different intensities that have no clear structure: The BwO is a permanent becoming, a developing, a life before the formation of established structures, opinions, meanings, and interpretations. A psychedelic experience can be described as a perception of the molecular, because psychedelic perceptions refer to the infinitely small, they are infinitesimal, and an identitary consciousness as father, mother, etc. is irrelevant. An essential element of this pharmaco-analysis is the

critique of bias within the analysis of drug experiences, e.g., that psychological theories are themselves part of a molar organization because they assume certain determinants of drug use that result from maldevelopments in the psychosexual stages, in unmastered challenges in coping with life [21], or in attachment problems with parents [22]. According to Deleuze/Guattari, especially in questions about drug experiences, deterministic theories cannot develop adequate conceptions of how substances affect the body and mind or are subjectively experienced, because patterns of justification are themselves shaped by the determination of the molar apparatus of meaning (this or that drug is dangerous because it produces hallucinations, i.e., no real experiences, disintegrates the family, etc., this or that drug is not dangerous because it maintains the family order or produces socially productive behavior). Determinism here can be described as an idea that assumes that events are completely dependent on pre-existing causes. For example, the cause of drug use is seen as a problem in early parent-child attachment or as a developmental disorder in the psychosexual development. The pharmaco-analysis emphasizes more the idea of indeterminism, i.e., the view that events are not caused deterministically, but rather occur multicausally or unilaterally. On the one hand, a pharmaco-analysis has such deterministic resp. ideological discourse productions in view. On the other hand, it observes not only how substances decompose or recompose the body and mind on the molecular level, but also how substances enter a society, produce discourse positions, reinforce relationships, dissolve them, and develop them [20] (p. 286ff).

In this sense, for an additional procedure, this pharmaco-analysis offers a framework to look at the molecular experience of drug users. The analysis of the molecular experience has the challenge not to create premature epistemological doublets [23] towards already existing discourse positions.

For this reason, François Laruelle's concept of non-philosophy and philo-fictions are used in the following. At the heart of Laruelle's non-philosophy is the assumption that a phenomenon cannot be grasped through existing interpretations. Existing interpretations and assumptions lead to epistemological duplications. In particular, if existing epistemological ideas are transferred to a phenomenon, then copies and duplicates of existing assumptions are created. Epistemological duplications turn the phenomenon into a clone of reality, into a fetishistic realism. For instance, in the photorealism of passport photos, product photos (fashion, food, cars), etc., representations of the world are created which function as copies of reality. These copies help to organize the perception of reality in such a way that it becomes an incontestable fixed idea, because images take the place of the real. With such an onto-photo-logical perception or an epistemological duplication, one observes the real itself through a photograph—not the object itself, but a representation of an identity. Therefore, epistemological duplications have a symbolic dimension: they are based on memories that classify the seen and make it identical with reality. This in turn leads to the problem that we become incapable of discovering new things because we are trying to discover the familiar in the unknown. The same can be applied to other sensory perceptions. By means of philo-fictions, a life of its own can be discovered in phenomena, independent of common scientific or philosophical assumptions. In particular, an interpretation of phenomena through philo-fictions means a rejection of already existing epistemological foundations [24] (p. 67). Simply explained, for example, with a materialistic, psychoanalytical or theological epistemology, I do recognize materialistic, psychoanalytical or theological content. Or if I pursue moral ideas, I discover in drug use a decomposition of moral ideas, wherein the drug abuse decomposes family or society. Philo-fictions counter this cognitive bias by substantiating epistemological reasons not according to the principle of "right" or "wrong", but according to the principle of "as-if". In order to find out what makes substances effective, specific drug experiences are investigated with non-philosophical experiments [25] (p. 235). Here, non-philosophy is to be understood as a practice that has its own non-autopositional rules (the suspension of philosophical authority) to develop a chaotic universe of multiple as-if representations (philo-fictions) [25] (p. 99) (see also [26]).

Important to this approach, as with Deleuze and Guattari, is the assumption that a phenomenon cannot be grasped through pre-existing interpretations. Furthermore, the point of philo-fictions is not to develop a copy or duplication of experiences, terms, concepts, etc. by means of ontological distinctions or aesthetic terms, but to expand the epistemological conceptual apparatus by means of hyperspeculations and fictive characterizations [25] (p. 239f). To set up experiments with hypothetical assumptions and conjectures that go beyond experiential reality is to establish the inherent life of phenomena independent of scientifically or philosophically common assumptions [25] (p. 22f). Such a turn away from identifications, or such a non-standard aesthetic of phenomena, means at the same time turning away from justifications of what is perceived, because cognition is traced back to existing coding. This allows phenomena to be de-subjectified and become a life of its own [27,28].

To produce a non-ontological cognition, the as-if representation, experimentation with fictions matters. This is also to be understood in analogy to science fiction literature. Laruelle transfers the principle of science fiction literature to philosophy, in order to separate science fiction from pure literature [27] (p. 488ff). Since fictional statements are neither true nor false, theses and antitheses are conceived as developmental possibilities in the form of as-if indeterminacies. The results of the synthesis thus remain open, as they can always be de-subjectified and subtracted in the sense of philo-fictions [24] (p. 132ff). An experimentation with philo-fictions does not verify or falsify the phenomenon but transcends it by preserving the identity of the hypothesis and preventing it from referring to existing philosophical assumptions. I.e. philo-fictions will not be supported in the last instance by an ultimate (positivist, critical, realist, idealist, absolute, etc.) insight [28] (p. 60f). In particular, instead of translating insights about phenomena into philosophical analogies, the experimental, non-philosophical work does not interfere with a particular scientific theory but models it in favor of the principle of internal self-similarity, in which recurrent structures and objects of fractalization emerge, generating an open ensemble, a universe of fractal epistemology. Methodologically, therefore, philo-fictions are to be applied via hyperspeculations [25] (p. 230). Hyperspeculations have the task of producing philosophical variables and indeterminants. [28] (p. 69ff).

Conclusion: At the core of Laruelle's non-philosophy is the assumption not to assign a phenomenon with the real by means of prefabricated conceptions, but to develop epistemological duplications by ontological distinctions or aesthetic concepts, and thus to let the phenomenon become a clone of reality. Rather, the reversal of this order should be made possible: in speculative philosophical experimentation with the phenomenon, a relative reality can be established. Thus, there is a chance for a phenomenon to develop a life of its own [24] (p. 22ff). A rejection of a cause–effect interpretation of the phenomenon means at the same time a rejection of the justification of what is perceived, because, as Gadamer suggests, "domination makes us deaf to the thing that speaks within its tradition" [23] (p. 274, translated by the author). Philo-fiction as a generic science of hyperspeculations does not produce the specific, but hypotheses which remain philosophically indeterminate as unilateralizations [25] (p., 69ff, 97f, p. 140ff). Philo-fictions are therefore to be understood as the practice that:

1.  Perform theoretical operations to produce hyperspeculations.

    For example, philo-fictions are a provisional tool, since psychedelic experiences, which exist in the mind of the drug user, cannot be answered with existing logical means.

2.  Develop their own non-autopositional rules of interpretation.

    For example, philo-fictions defend speculative thinking as a way of achieving new insights and theories. However, for them to be accepted as "scientific", they must be also critically examined.

3.  Assume multiple "as-if" representations.

For example, philo-fictions can be used to make claims that lack a rational basis. However, these claims could be proven wrong or right in the future through appropriate investigations and studies.

Now, in order to find out how the psychedelic experience, the molecular perception takes place by means of a non-philosophical framing through philo-fiction, psychedelic experiences of 5-MeO-DMT are investigated (Section 2). Subsequently, as suggested by Laruelle, psychedelic phenomena are viewed through distance and framed within hyper-speculations [24,28] (Section 3). Finally, these hyperspeculations will provide an open synthesis for the question of how to understand experiences by taking entheogens beyond the usual epistemological narratives (Section 4).

## 2. Phenomenon of 5-MeO-DMT Experience

Many scientific research reports, self-experiments, and experience reports of the phenomenon of a 5-MeO-DMT experience are contained in the largest psychedelic databases of the Multidisciplinary Association for Psychedelic Studies (MAPS) as well as the Erowid Center. Hundreds of similar procedures are described here. Some exemplary excerpts from 5-Meo DMT experiences are included below. These reports can be seen as an introduction to the phenomenon of the 5-MeO-DMT experience (the main report for this article will follow these short descriptions). In all cases, 5-MeO-DMT was vaporized and inhaled or injected:

> "Intense, very short-acting period where shifting dark and light geometric patterns are seen everywhere in my visual field and somehow my body seemed to be incorporated with them. I had the sense of being in two places at one time. I could still see/relate to my bedroom, but I also felt like I was in 'no space'" [29].

> "... I had the feeling, a visualization of being part of the universe of beings, all active in our daily, interwoven tasks, still moving at an incredible rate, and with a longing for a single group/organism awareness and transcendence..." [30].

> "Within thirty seconds... brief collapse of the EGO, and loss of the space–time continuum" [31].

> "I realized I hadn't felt such fear ever before in my life. It was true terror, but something made me feel comfortable. There was no way out... Gradually I was feeling cold and transported elsewhere, I felt my body was succumbing and my mind had been projected out of it into a void, some sort of space where my perception of time was no longer working or reliable. Everything physical felt meaningless, there was true force there, a power the likes of something I had never seen before. I started having flashbacks, thoughts of my life, the troubles and destruction I caused to the lives of different people... I became suddenly convinced this was an opportunity to reconsider several aspects of this all, and meditate over what was valuable and truly important in life" [32].

Field reports collected by Ralph Metzner, psychotherapist, and former professor of psychology at the California Institute of Integral Studies in San Francisco, on exploring 5-MeO-DMT also describe qualitatively similar experiences:

> "... Although I was in a room, I could feel the sensation of earth, the dry desert floor... Strong visual hallucinations in Orb-like brilliance, diamond patterns moving wave-like through my visual field" [33].

The 5-MeO-DMT self-experiments of Alexander Shulgin, chemist and pharmacologist who became known for the systematic development of synthetic tryptamines (e.g., DMT) and phenethylamines (e.g., MDMA) [34], described dissociative episodes:

> "I was not present in my body or in time... my consciousness was completely without reference frame... and in my head it was screaming, 'Shit! You killed yourself.' I repeated this a few times, in supreme agony... I concentrated on breathing, and that helped me to survive (mentally)" [34] (p. 533f).

The accounts of Stanislav Grof, the founder of transpersonal psychology, also describe similar experiences:

> "I lost all contact with the world surrounding me, which disappeared completely (. . .) I had no concepts, no categories for what I was experiencing. (. . .) I became consciousness that encountered the Absolute. It possessed the brightness of myriads of suns, (. . .) It seemed to be pure consciousness, intelligence, and creative energy that transcended all polarities. . ." [35]. (pp. 251–257)

Ralph Metzner, who had been researching LSD and psychotherapy since the 1960s with Timothey Leary and Ram Dass, among others, experienced 5-MeO-DMT as follows:

> "A shattering annihilation, a feeling of finding myself inside an explosion and disintegrating into countless little pieces. (. . .) There was a feeling of being in the core of the psyche, an awareness of 'All and Everything' and at the same time 'This is THAT'. (. . .) unimaginable ecstasy. . . a swirling, interweaving grid of light patterns everywhere and all around. . . Then this kaleidoscopic pattern-field dissolved my body into itself so that I no longer saw it—I had become a part of it. (. . .) The weaving, waving field of geometric shapes and lines folds and falls down on me (. . .). These nets form my body, accumulate in certain areas and form organs like my eyes. They also form all the other bodies and shapes around me. Each individual is a kind of accumulation in this infinite, ever-changing molecular web. Every thought, feeling, or experience is also a local accumulation in this holographic matrix of all possibilities" [9] (pp. 41–47).

To interpret these descriptions of aspects of molecular perception and levels by means of a hermeneutic method instead of by philo-fictions would mean to pursue the distinction between the latent sense structure and subjective sense representation [36] (p. 241). To this end, it would be possible to follow in the traces of Grof, for example, who retrospectively describes the experience in that he believed he experienced the Bardo (the tibetian intermediate state before rebirth into the next incarnation) or believed that he experienced the "Dharmakaya, the primal Clear Light that, according to the Tibetan Book of the Dead, appears at the moment of our death" [35] (p. 251). Also helpful would be the reflections of Metzner, who places his experience within themes of Tantra and Kundalini Yoga [9] (p. 42). However, this would lead in the context of a pharmaco-analysis and philo-fictions to analytical and conceptual duplications. The analysis would lead precisely to theological, spiritual, or anthropological traces and would run the risk of reproducing epistemologically familiar concepts, despite the clear analogies to themes in the world of religious or spiritual practices.

Therefore, I refer to my own observed experiences, in order to subsequently create hypotheses for the interpretation of the 5-MeO-DMT experience. Due to my own research interests, I was in San Francisco in 2022 and received an invitation to be present during the medication of 5-MeO-DMT in a professional setting. The small group included the provider who gave the 5-MeO-DMT, a counselor who supported the participants directly before and after the treatment, and six participants. Among them was M., who was able to free himself from his long-standing opiate addiction through the support of recurring 5-MeO-DMT sessions. For him, he explained that the way out of addiction had been long and connected with setbacks, but the "medicine" had shown him the way.

The following explicit report, by another person of this group, is especially authorized for this text. This report also serves in the following article to make hyperspeculations. In this report only the immediate sensory impressions are presented. Conceptual interpretations or identifications of the person are excluded:

> P: "It's almost impossible to talk about that. You only have words to describe what was there. It is beyond anything I have ever experienced. After exhaling, the shapes of the objects turned into patterns, started to move, to run into each other, to rotate. Geometric figures, fractals emerged, flowing into each other. This flowing into each other became faster and faster and then I was catapulted into

nothingness. At a tremendous speed, I was suddenly in an infinitely large black space, as if in a vacuum. I can remember that it was hard to breathe and that I was gasping for air. It was frightening and fascinating at the same time. There was no time to think about what was happening. It was like my ego was dissolving, it was no longer me as a person perceiving all this. The space in which I was like outer space, nothing to hold on to, to touch or anything familiar. Maybe all this lasted only a few seconds, but it was as if time had stopped. A very clear moment of no more being on earth and having arrived in the absolute, boundless nothingness. (. . .) No more I, no mind there that could grasp anything. Then a feeling of happiness sets in, after I somehow realized that I don't need to be afraid of this state, that even if I don't come back, it's good here in the nothingness. (. . .) Slowly my senses came back too, but only for a short moment, I heard music, chants (. . .) for a short time, then I was able to move in this vacuum, I was pulled to holes, tunnels, wavy surfaces, I flew through extraterrestrial canyons on a planet or maybe in atoms, in microscopic worlds. It was big and small at the same time. I, It, everything around me had no more dimensions. It was a flight through micro- and macrocosm. Everything in a black background. Not a frightening black, but a black that envelops you. Exiting a tunnel, I was in a desert. It was bright and pleasant. The flight had stopped. It was like there was another station here. I saw very bright light and a kind of sun, then an eclipse, the flight continued, there were several stations, and I had the feeling that I was experiencing backwards what I have experienced so far in my life (. . .). Then I felt a great heat in a body. I became hot, unimaginably hot. Then rays shined from my body into the sky. I had the feeling that I was stretching my hands with the rays into the sky. Then I heard a voice: You are a God now. I said to myself that I cannot be a God at all because everything around me is like God. (. . .) Then a kind of dialogue with the vacuum, with the microcosm or macrocosm or with nothingness began. Anyway, it was not a monologue. I wasn't talking to myself, but there was something talking to me, but not in a language as in a discussion, but it was a mediation of a feeling or an exchange of feelings, in any case a speechless understanding, with the realization that everything is connected to everything, that in reality there is no separation between me and my environment, that we are all connected, not as a kinship, but as a living organism, animals and plants, also the non-living, stones, the desert, the sky (. . .). I got a feeling of indescribable happiness that I had to start crying. Slowly, my ego also came back, I could perceive myself as a person again and then, however, I was brought back to the state of happiness and had to cry again because the knowledge that everything is connected was the most beautiful thing I have experienced so far. I wanted to keep that feeling and take it with me. That I am part of you, and you of me, we all together only make a whole. I can't put it all into words, anyway I had had an unprecedented feeling of gratitude and connectedness to my life and the world. The place I was is still there in me. The feeling of being connected is still there and I am afraid that this feeling will disappear again. I can't believe it all lasted just a few minutes, it felt like forever".

Now, to prepare this fantastic narrative for philo-fictions, in the next section hypotheses will be developed. Therefore, the next section describes the exegetical order of a non-philosophical philo-fiction, followed by the first hypotheses for speculations about the described 5-MeO-DMT experience.

## 3. Philo-Fiction: Hypotheses of the 5-MeO-DMT Experience

In the following, hypotheses are derived from the described psychedelic experience. With these hypotheses, philo-fictions are subsequently generated to remove ontological monocausalities. Finally, this should make it possible to develop speculations about

psychedelic experiences with the aim of transcending previous interpretations in such a way that further epistemological ideas about a psychedelic experience can be formulated.

- Hypotheses

In view of the report of the experience and the description of images, spaces, and interactions that took place during the experience of the report of P., three action-theoretical or praxeological levels of the experience can be identified:

1. Seeing: Perception of hallucinations as geometric patterns, fractals, obscuration, and white light.
2. Sense of space: Experience of a tunnel or breakthrough into another world. Three- or higher-dimensional space. Simultaneity of micro-macrocosm. Feeling of oneness, all-oneness, and consciousness.
3. Information exchange: Communication, exchange, and contact with entities.

There are already existing discourses and positions on these levels of the experience, which can be further developed as hypothesis:

1. Materialist-reductionist, neuropsychological hypothesis: the experiences and images are hallucinations. Whether what is perceived is real or not is irrelevant, as the experience cannot be verified nor falsified, as it is subjective [37].
2. Hypothesis of other worlds: drug experiences provide access into an n-dimensional hyperspace. Since everyday perception minimizes spatial orientation, psychedelics can entangle access to otherworldly dimensions of space and time and provide access into these fields [38,39].
3. Transpersonal hypothesis: the sensations transport transpersonal contents. The communicating beings merely appear alien because they represent unknown aspects of ourselves, are part of a collective unconscious, or embody unconscious archetypes (creators, angels, goblins, spirits, etc.) that speak to us [40,41].

These positions will now be taken up in the following to develop three philo-fictions.

- To (1) Philo-fiction about the materialist-reductionist, neuropsychological position

The state of pharmacological research on the effect of DMT in the brain can be summarized by the following: DMT molecules activate receptors for neurotransmitters, such as serotonin, a neurotransmitter through which brain cells exchange signals. Currently, the sigma-1 receptor (Sig1R) seems to be of importance as an explanation of the pharmacological DMT experience. This receptor is evolutionarily more closely related to the fungal enzyme sterol isomerase than to mammalian neurotransmitter receptors (see also philo-fiction 3). On the membrane, it is able to interact with other neurotransmitter receptors and to change their function. Within the cell, this molecule is supposed to bind to anti-stress proteins, to activate the production of anti-inflammatory molecules, and to deactivate epigenetic mechanisms in the cell nucleus, such as memory-related regions of processing negative emotions and sad memories. This means that DMT molecules interact with multiple receptors that relay signals across cell membranes, within the cell, or in the nucleus, triggering a wide range of biochemical signaling cascades and stimulations. Strikingly, brain regions communicate with each other that have little connection in everyday life [42,43]. In relation to hallucinations, it is assumed that they arise due to the spontaneous discharge of remaining intact neurons and trigger a dual system experience of extreme contrasts, such as light and dark [44]. Bresslof et al. conclude in their study "Geometric visual hallucinations, Euclidean symmetry and the functional architecture of striate cortex" that hallucinations are based on perceiving neuronal connection patterns as visible fractals, funnels, tunnels, phosphenes between retina, striate cortex, and neuronal circuits. But these patterns of connections can be seen in dark rooms as well as by blind individuals, without memories of an image-forming world [45].

Hypothetically, seeing fractals, funnels, tunnels, and especially phosphenes could also be understood as an image of the body interior because the patterns remain stable despite

eye movements. This leads to the speculation that they are not produced in the eyes, by the eyes, or in front of the eyes, but in retinotopy. Psychedelic hallucinations would accordingly be an inside view resp., an inside insight, or a direct consciousness contact to the brain and body insight.

By means of the new-materialist position of flat ontologies [46], further speculations can be developed in this regard: The assumption that matter is not a passive substance brought into existence only by the intentionality of the observer, but by co-construction [47] or intra-action, means that an agency of matter and the existence of independent entities can be assumed [48]. With this assumption, it is possible to further speculate about a flat ontology as a model for a reality in which substances, even imagined ones, do not stand in hierarchical relations to each other [49]. Phenomenologically, the perception of fractals, funnels, tunnels, and phosphenes is interesting in this regard, as they can be understood as intra-agents [50], which are ontologically entangled with the perceivers [24] (p. 144ff). Once more, this leads to the speculation of a direct perceived connection with body parts.

With such a conception of molecular materializations, the real appears not as an "onto-photo-logical appropriation" [24] (p. 55ff), i.e., not as a picture-like copy of the external world, but as an agential incision in space and time. The agential cut decides within the ontological (and semantic) indeterminacy inherent in the phenomenon [50]. This suspension of onto-photo-logical perception leads to the elimination of the pre-reflexive classification of what is seen, which classifies the observation and makes it identical with reality [51]. The result of a non-identical observation and a direct perceived connection with the body enables the recognition of the new, in that the new itself becomes a process of signification. As a result, the subject and object, the observer and the objects that were seen during the drug experience can no longer be clearly separated. The observers are part of the observable, and the observable is part of the observers.

With these speculations, the first further hypothesis can be derived:

> Whether hallucinations are true or not is indeed irrelevant. However, not to doubt the subjective experience, but to contextualize the described neuropsychological position in terms of an action theory: Because what is seen is relevant to the perceivers, the associated experiences lead to a re-coding of perception by leading the subjective received cognitions to a cellular or molecular level of themselves or an agency in themselves.

In this regard, it would be worth considering further whether it is precisely this feature of the 5-MeO-DMT experience, the experience of non-knowing, non-identity, non-being-identical, and non-me, as well as the directional action of substance and the sustaining of the shock of the senses, that gives birth to a new agentive self. Pursuing this hypothesis would require addressing the questions of whether a distinction between reality and appearance is abolished and whether an experience remains ascertainable both empirically and as a difference between the representing act and represented content. With this discussion, it would be conceivable to ascribe subjective meaning to the content of experience, as well as to process insights and knowledge of experience over the course of time [52].

- To (2) Philo-fiction about the other worlds position

In the psychological discussion about space perception, a kinesthetic, acoustic, and visual construction of a three-dimensional space is assumed. Fundamental for this is the individual location of the person in space resp. perception of the distance of objects from the observer, i.e., spatial orientation is dominated by depth perception. Nevertheless, spaces can be described mathematically and physically, which are not accessible by everyday perception:

On the one hand, in "Ueber die Hypothesen, welche der Geometrie zu Grunde liegen", Bernhard Riemann describes in 1854 how non-Euclidean geometries become transcendable by $n$-fold expansions of size in any number of dimensions (see, e.g., Tesseract, four-dimensional hypercube) [53]. This Riemannian multiplicity formed an essential reference point for the development of general relativity, theories of the space–time continuum, as

well as multidimensional hyperspace. The idea of a hyperspace has preoccupied both physics in order to develop speculations to reject the relativistically-based impossibility of assuming parallel universes [54]. On the other hand, mathematical alternative models, based on string theory or quantum mechanics, propose a two-dimensional ordering of the universe under the concept of a holographic universe. A hypothesis by Leonard Susskind suggests that the three- or more-dimensional reality is a projection of information stored on a two-dimensional surface at the edge of the universe [55]. The 2D models are again based on speculations of the quantum physicist David Bohm and his quantum cosmology, which is related to the many-worlds interpretation (multiverse and a multiplicity of dimensions). Bohm's investigations show that the cosmos does not comprise elementary particles, but instead it is an unbroken, undivided whole, in that each element, carries the information of all elements as an implicit order on a universal field. Reality generates itself as a holomovement, a holographic web, through an implicate order [56].

In this context, the experiments of biologist Rupert Sheldrake [57] allow for further speculation as to whether these dimensions could also be found in the human or animal body. Sheldrake argues that cells, tissues, organs, organisms, even inorganic substances contain their own information for form formation (morphogenesis). These developmental programs of form formation resp. information fields (morphogenetic fields) are determined from forms of similar organisms of the past. A creature that is developing is therefore in morphogenetic resonance with countless earlier creatures, ultimately all creatures are then parts of that One field, and can perceive and feel the morphogenetic resonance of the other parts [57] (p. 145ff).

In this context, psychiatrist Luc Ciompi also describes connections of macro- and microstructures. His fractal logic of affect is based on the idea that emotion and cognition interact in a regular way in all mental performances. In particular, emotional tensions can lead to nonlinear bifurcations or to jumps into other patterns of feeling, thinking, and behavior. These interactions between thinking and behavior are self-similar on any individual and collective level, i.e., the smallest effects of feelings are to be rediscovered in time-permanent culture or time-specific ways of thinking in society: "The greatest in the smallest, the smallest in the greatest—infinitely modified" [58] (p. 128).

Hypothetically, in the context of the issue raised for this article, the ancient notion of the microcosm–macrocosm analogy resp., the speculation of the One field which connected everything could be taken up, which postulates that the world as a whole is in a similarity relationship to the microcosm and the structure of the whole is repeated in the small and vice versa. The principle of cognition is based on the fact that cognition in the small involves cognition of the large [59]. The consciousness researcher Terence McKenna argues further in this regard that the body itself should be thought of as a holographic object that is itself part of a hyperdimensional matrix of macrocosm and microcosm and that DMT would be the "spaceship" [60] (p. 93) to explore new dimensions of immanence and other worlds due to its flat ontology.

Following on from this, with Markus Gabriel's concept of the sense field ontology [61] (it could be argued that there are no flat ontologies and pure levels of immanence, because by the functional ontological difference between sense fields) it is assumed that objects are mutually conditional, and objects are individuated with functional specifications [61] (§ 8). This premise can lead to a sensory epistemology since everything that exists appears in a sense field.

In this respect, what is seen on a 5-MeO-DMT experience can be used to develop a spatial sense field ontology that mutually presupposes the functional ontological difference between sense fields and objects and individuates objects as a functional specification. Thus, it would be hypothetically possible to explore 5-MeO-DMT worlds by means of methods of sensory cognition.

With these speculations, the second advanced hypothesis can be derived:

If the 5-MeO-DMT experience produces a cosmological reality that extends beyond space and time, and which ultimately represents projections from other

dimensions or a deeper order of being, then its ontology can be discerned in the experience.

Whether the universe experienced in 5-MeO-DMT experiences is described in models as two-dimensional, multidimensional, or fractal is irrelevant to describing the experience of another world; perhaps the world is indeed only a slice within a larger multiverse. Nonetheless, there would be a fundamental invariant common to all worlds, which is to recognize it [52]. Since 5-MeO-DMT leads to experiencing multiple worlds, the question would rather be with which models a sensory cognition of these 5-MeO-DMT worlds becomes possible. With a materialistic approach, controlled experiments would have to be conducted in the individual DMT "parallel worlds," i.e., it would be necessary to send "psychonauts" into these worlds, these worlds would have to be measured and mapped, geographical images would have to be created, which in turn would allow for the description of structures. Qualitative and quantitative comparative studies would have to be carried out, which, while considering a questioning of the disciplinary framings that prevail the solipsistic closures of value universes [62,63], would still have to be critical of inductive conclusions.

- To (3) Philo-fiction about the transpersonal position

For the transpersonal position and its speculations regarding the archetypes transcending through DMT in the form of God, angels, jesters, etc., depth psychology or transpersonal psychology can be taken as a reference model, because it is able to develop a causality for affective phenomena [18]. Here, the focus is on the hypothesis that in the "DMT dream" the alien (masquerading as a God, entity, or extraterrestrial being) emerges and draws attention to individual deficits or fears. The therapeutic goal would be to reconcile with the alien and begin to heal a psychological discontinuity [40] (p. 43). In the transpersonal position, the drug experience forms a transcendence level in which information is processed [64].

With Marshall McLuhan's communication theory, a different kind of speculation arises in relation to the processing of information. His thesis is that the form of a medium embeds itself in the message, creating a symbiotic relationship between the medium and message. As a result, the medium influences the perception of the message [65]. Applied to the substance DMT, this would mean that DMT itself contains information. Relatively, the active ingredient tryptamine—which induces the drug experience and which occurs naturally in fungi (psilocybin; 4-phosphoryloxy-*N*,*N*-dimethyltryptamine), plants (leguminous plants; *N*,*N*-dimethyltryptamine), and animals (toad secretion; 5-methoxy-*N*,*N*-dimethyltryptamine)—as media itself, that means fungi, plants, animals, contains a message or has a massage [65]. Therefore, McKenna hypothesizes, based on the pan- or trans-sperm hypothesis [66], that tryptamines arrived on planet earth through spores from space, via meteorites or comets. This astrobiological hypothesis states that simple life forms move through the universe over great distances by means of light pressure from the sun, through meteoroids or gravitational fields, and brought beginnings of life to earth. Fungal spores and other prebiotic organic matter can survive the vacuum and cold of space [66]. McKenna argued that the Homo erectus consumed psilocybin mushrooms, and that the evolution of Homo erectus to Homo sapiens and the self-awareness and consciousness was driven by the consumption of 4-phosphoryloxy-*N*,*N*-dimethyltryptamine. McKenna also suggests that the evolution of abstract thinking and language may have been caused in the context of tryptamine synesthesia. Moreover, McKenna asks in this context whether the medium involves a message from outer space or whether actual contact with aliens occurs in the psychedelic experience through the medium of 4-phosphoryloxy-*N*,*N*-dimethyltryptamine [40] (p. 143ff).

In reference to DMT, which McKenna himself consumed, he described encountering entities as follows: "I encounter 'self-transforming elf machines' that are creatures, entities perhaps, although they are not made of matter but of, as far as I can figure out, syntax-driven light. (. . .) They use a language that you can actually seen, it's made of sound (. . .). And the whole point of the encounter, from their point of view, is to teach you to use

this" [65]. To this end, McKenna further speculated that it is necessary to find ways to remain in the DMT world longer and learn this language. Visionary artist Allyson Grey has also experienced alien language in psychedelic sessions. For this, she has developed a "secret writing alphabet": "The Secret Writing in my art is my interpretation of the occult symbol system I saw on a mystic journey in 1971 at the age of 19. Secret Writing has now been reported by many psychonauts. Of the infinite variety of symbols I witnessed washing over all surfaces and wafting through the air, I selected twenty letters, an alphabet I placed in a chosen order. Personally, I feel that my letters are meant to be untranslatable" [67]. However, should the medium contain a message, it would be impossible to learn an alien language naturally, according to Noam Chomsky, who has spoken hypothetically about alien language, because human language is subject to a genetically predetermined universal grammar. If at all, humans would have to decode an alien language by way of scientific analysis [68].

With these speculations, the third continuing hypothesis can be derived:

> If an information exchange can be established with fungi, plants, and animals through the medium of tryptamines, the form and content of communication will not allow the collective unconscious to become conscious, but boundaries of speciesistic information processing, grammar, and communication.

In order not to dock this hypothesis to a discussion of an anti-speciesist use of language [69], with a view to a more advanced transpersonal position, the unconscious itself would have to appear as a perceived object of desire. The unconscious would thus no longer be the hidden principle of a transcendent plan of organization, but would appear as part of an immanent level of consistency that refers to itself during its construction: The drug gives the unconscious the immanence [25] (p. 386f). Here, it would be necessary to ask further how the immanence of the world—as "a unilateral semi-mysticism that the world needs only to combat realism and determinism" [24] (p. 153)—should be further de-coded, as well.

## 4. Open Synthesis

For the rationale of why an experience of 5-MeO-DMT might be an antagonist of depression, the described speculations provide a few starting points to summarize the pharmaco-analysis:

- (1) Open synthesis of the materialist hypothesis

The first hypothesis describes the following starting point: In clinical and pharmacological studies, the effect of psychedelics on the organism are being decoded in order to isolate more specific mechanisms of efficacy to control the corresponding tolerability, dosages, and side effects. One possible mechanism for the effectiveness of psychedelic substances against depression is their ability to influence the activity of neurons in the brain, e.g., by reducing the default mode network (DMN) and increasing neuronal plasticity. Increased activity of the DMN is often associated with depression and anxiety disorders. If the DMN is active, the brain does not react to external stimuli, learns little new information (limited neuronal plasticity), and instead produces inner thoughts and ruminations that are tied to past (negative) experiences. Pharmacological studies have also shown that psychedelic substances have an influence on serotonin levels and affect the activity of serotonin receptors (5-HT2A receptors) in the brain. Serotonin is a neurotransmitter involved in many processes in the brain, including mood, anxiety, sleep, and appetite. A lack of serotonin is often associated with depression [4–7,70]. However, tricyclic antidepressants such as Prozac, etc. also have an effect on serotonin levels, and it is fundamentally controversial "whether the fluctuations in serotonin levels are responsible for the antidepressant effect". At the very least, there is "no clear connection between serotonin levels and depressive symptoms" [13] (translated by the author, p. 236). In particular, chemical balancing as the only explanation of why psychedelic substances are effective against depression would fall short of the current state of research. This conclusion is also supported by Borbély [71]. He describes

further that in contrast to traditional antidepressants, psychedelics have an immediate and enduring therapeutic effect which is correlated with mystical-type experiences. Further studies have to focus on mystical factors or the altered state of consciousness rather than on the search for neural correlates alone. Also, Sjöstedt-Hughes shows in his study "On the need for metaphysics in psychedelic therapy and research" [72], that there is evidence of an afterglow phase following psychedelic experiences, in which participants are open to new ideas about themselves and their reality. Therefore, it seems more important that, with the help of psychotherapists and social workers but also philosophers, that a metaphysical processing, and a life situation specific integration of the experience into the everyday life of participants should take place to help them overcome depression.

- (2) Open synthesis for the other world hypothesis

The second hypothesis started with the following argument: Drug user experiences describe a breakthrough into another dimension, the simultaneity of micro-macrocosm, feelings of all-oneness, and consciousness. This raises the question of what consciousness actually is. Can consciousness be described by using physical, chemical, or molecular biological analogies? Prentner [73] investigated this question and found that the most popular idea by neurobiological or functional theories of consciousness is that consciousness is a (structural or systemic) property of information-processing networks. He assumed that either neuroscience or fundamental physics is the main relevant discipline which can contribute to a science-based understanding of consciousness. But he suggested that molecular sciences with a pan psychic turn, i.e., the idea that consciousness is irreducible and ubiquitous in the universe and mind or a mindlike aspect is a fundamental feature of reality, could help to understand the molecular mechanisms related to the phenomenology of consciousness [73] (p. 5).

The description from the 5-MeO-DMT report could support the pan psychic synthesis as well as the hypothesis of Deleuze/Guattari, that molecular consciousness crosses, overlaps, and transforms molar consciousness, so that a new powerful self can emerge that becomes aware of its life situation and can transcend depression. Conversely, if we look at how depression works, it may be easier to understand why psychedelic experiences of feeling connected to everything can act as an antagonist to depression:

The feeling of depression restricts people's behavior to such an extent that they settle into their depression. People brood over problems but do not tackle them, offers of help are not accepted or structurally damaging living conditions (poverty, stress, time pressure, bullying, lack of work-life balance measures, etc.) are reinterpreted as individual shortcomings because people do not want to come into conflict with their environment. The result is that depression becomes a permanent condition over time and consequently reinforces the state of being unable to act. The success of depression is based on a lost connection to a conflict with stressors and the pursuit of individual desires [2]. With regard to the mechanisms of psychedelic substances—be it the neuronal ability to deal with new content and the psychedelic-induced stop of worrying or the mystical experience of being part of the world again and the benefit of being able to master difficult situations—it can be said that precisely those moments of new views, experiences of awareness, all-oneness, of a different self in a different time and space are effective counterparts of the depression. Athletes are also familiar with the unity of body-mind-environment, the feeling of being one with everything. They describe this as a state of flow. The difference between athletes and people who feel depressed is that depressed people do not primarily challenge their body, mind, or environment. But with psychedelic substances they do, and they train their self-awareness and self-efficacy.

- (3) Open synthesis for the transpersonal hypothesis

The third hypothesis started with the following argument: During the psychedelic experience, an information exchange, communication, and contact with entities took place. In analytical psychology, the confrontation with the other, masked as an alien, God, monster, etc., indicates that psychedelic experiences deal with fears of failure and feelings of shame,

that psychedelic experiences make wishes come true and that the compensation of the one-sidedness of a person's life arises in order to "become whole again", as Jung described [74] (§ 440, 445). In particular, depression is not the product of a malfunctioning brain, but of a malfunctioning way of thinking, e.g., that life should be without difficulties or that depression is a symptom of not dealing with these difficulties or shadows. Jung suggested that his patients should face their shadows, the problems of the present, in order to initiate the process of self-transformation. In this sense, the psychedelic experience could lead to the reawakening of what has slipped into the unconscious, into the molecular, i.e., the qualities that stimulate human existence.

A subject-oriented turn to this assumes that substance use has a benefit for the user; "because it helps", "it shows the way", etc. and therefore takes place as a purposeful action. In particular, the effect of psychedelic is based on a self-therapeutic, self-medicating benefit of drug use, e.g., to be able to cope with difficult life situations or depression, to find space to reflect on the meaning of life, or to be able to look at problems from a different perspective. Here the question is, how the insights gained from psychedelic experiences can be developed as self-efficacy expectations through integration into the everyday life. Again, as Sjöstedt-Hughes suggests [72], an additional metaphysic integration into a psychedelic-assisted psychotherapy can be more effective, if the participant's sense of themselves and the reality in which they exist would be part of it: Because the experience could be more comprehensively framed, could be less delusional once a participant realizes that metaphysical positions amplify their experience and the therapeutic efficacy.

In conclusion, it is worth considering that these very characteristics of the psychedelic experience: (1) the neuronal pausing of molar thoughts, (2) the ecstatic experience of being whole again and part of the world through connecting with everything, and (3) the confrontation with painful contents that are masked by the alien are the very characteristics that allow for a renewed, more powerful, and effective self to emerge, a self that is able to overcome its depression. These characteristics also describe experiences of immanence because the psychedelic state changes the empirical appearance of the ego, which begins with the experience of the dissolution of ideas, identifications, and molar codes of reality.

With this access of awareness about the dissolving of identifications, philo-fictions can also be tied back to the questions raised in the introduction: Questions about the dimensions of experience as well as the materiality of experience. As described above, philo-fictions are a provisional tool as a way of achieving new insights and theories that lack a rational basis. To open up the field of further questions, it is worth considering that the described philo-fictions allow for speculations about different levels of reality as well as the agency of matter. One aspect stands out here. If in the psychedelic experience the object forms of the world are no longer in the foreground of experience, but rather the formless and unrecognizable, then a dissolution of boundaries with the self also arises. In this state of de-subjectification, there is not one absolute truth or one identity (e.g., as a depressed, drug-addicted, etc. person), but n-countable experiences can become n-countable identities [24] (p. 114; 76). The non-identical materializations can rise to ambiguous percepts, and the absence of molar organized object forms can lead to a variety of molecular layers that as a result cannot be grasped in a conceptual self-setting or synthesis, because the non-identical materializations cannot be reduced to any original matter or to any empirical reality or philosophical idea [75,76]. The philo-fictions refer to fractal dimensions of mind and matter, to the immanence of individual and objective existence. The individual does not charge himself with the divine spirit, but the de-subjectivized individual can feel part of the whole again through the experience of immanence. Therefore, the psychedelic experience is not an experience of spirituality in relation to an unaccountable transcendent reality. Further questions in the context of psychedelic research would be how immanence, as the understanding that all existence is already contained in the ego (consciousness), can be further explored. In the view of philo-fictions, a meaningful open synthesis for a pharmaco-analysis would be a transgression within the frame of entheogens. When Gordon Wasson, Richard Evans Schultes, and Jonathan Ott, among others, developed this neologism in

the late 1970s, this was carried out with the intention to describe psychoactive substances as entheogenic (ancient Greek "filled with God, blessed, coming into being") in order to describe the psychedelic effect in a sacred context which is linked to mystical experiences [3]. If the developed philo-fictions are now also understood as fictional ontologies, then the workaround notion of non-entheogens could be used to highlight the fractal dimensions of mind and matter, the non-identical, the experience of de-subjectification, and notions of immanence.

**Funding:** This research received no external funding.

**Institutional Review Board Statement:** Not applicable.

**Informed Consent Statement:** Informed consent was obtained from all subjects involved in the study.

**Data Availability Statement:** Data is unavailable due to privacy or ethical restrictions.

**Conflicts of Interest:** The author declares no conflicts of interest.

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
