# Peer review of "Pharmaco-Analysis of Psychedelics—Philo-Fictions about New Materialism, Quantum Mechanics, Information Science, and the Philosophy of Immanence"

_philosophies, doi:10.3390/philosophies9010007_

Round 1

Reviewer 1 Report

Comments and Suggestions for Authors

Taking into account that the psychedelics are of very interesting medicinal substances for the correction of human mental health in general sense. Thus, tryptamine (DMT and its derivatives) are of more potent among all psychedelics, it seems to me due to their effective affinity for receptors in the brain by analogy with serotonin. In this context, this article dedicated to the philosophical discussion on the relationship between man and psychedelic substance is an important and can be intest for readers of this iournal as well as useful for reasearchers who study psychoactive drugs.

I would like to recommend this article for the publication due to it observe an interest vision of psychedelics applications and their action human consciousness.

At the same time, I would like to suggest some small corrections:

1. Page 1, line 38. 5-MeO-DMT should be placed outside the parentheses in this sentence since it now appears that DMT and 5-MeO-DMT are the same substance.

I think that it will be more correctly: «However, the duration of DMT (N,N-dimethyltryptamine) or 5-MeO-DMT is much shorter (15-90 min) and seems to be more suitable for a broad use in psychotherapy.»

2. DMT and 5-MeO-DMT are mentioned in many places in the text of the manuscript, including, for example, in Part 3. "Philo-fiction: Hypotheses of the 5-MeO-DMT Experience" on page 6, I ask the author to check whether these substances are actually mentioned correctly everywhere because, as I have already noted in my question 1, these are two different compounds.

Reviewer 2 Report

Comments and Suggestions for Authors

This article is interesting because it embraces a very “trendy” topic (namely psychedelic research) from an unusual perspective and stresses, correctly in my opinion, the total neglect of any non-mechanistic explanations (“cause/effect narratives”). The alternative, if I understood the text correctly, is an account in terms of “philo-fictions” which work similarly to the way science fiction might enable us to transcend our current understanding of the (scientific) state of the world, namely as “as-if” statements, which are neither true nor false (that’s not their purpose). 

That said, I think their alternative proposal could be a bit clearer. I often encounter terms with should be better defined. In particular

-       What are "epistemological duplications"? 

-       Why Do Deleuze/Guattari use the names molar/molecular perception? 

-       What is a “deterministic theory”? That seems to be a non-standard use of deterministic here

My impression is that the author is on to something but I cannot really say what. The general methodology of “philo-fiction” put down at the end of section 1 should be revised better explained.

Further: Why is the position according to which “what subjects see is hallucinatory, veridicality is not important” called “materialistic/reductionist”? Anil Seth’s book Being You (2021) might be interesting in that respect, since it qualifies as materialistic and Seth speaks of “perception as controlled hallucination”. A materialistic (brain-based; mechanistic?) approach might lead to such a view, but a non-materialistic one might too in the end.

Overall, I am very sympathetic with the author’s diagnosis, but less with the treatment. I am not sure what the philo-fictions would enable a clinician to do specifically. My impression was that they are just alternative interpretations of what’s going on. But this certainly does not exhaust what the author wants to say.

Further comments/possible additions

There has been an excellent recent article by Alex Borbély, a long-time medical researcher and neuroscientist, with a similar complaint: psychedelics research is often carried out without stressing the importance of the first-personal (experiential) dimension vis-à-vis a materialist/mechanistic view of brain function.

A. Borbély, “From LSD to LSD – A Personal Trajectory”, Mind and Matter, 2022.

Throughout this article, nothing has been said about modern theories of consciousness. I found that omission surprising. On the one hand, this is understandable since most of these theories are materialistic/mechanistic. On the other hand, there are some interesting non-mechanistic models available. For example:

R. Prentner. “Consciousness: a molecular perspective”, Philosophies, 2018.

D. Hoffman et al. “Fusions of Consciousness”, Entropy 2023. 

P. Sjöstedt-Hughes. “On the need for metaphysics in psychedelic therapy and research”, Frontiers in Psychology, 2023.

Minor: l.44 This sentence sounds odd. 

Round 2

Reviewer 2 Report

Comments and Suggestions for Authors

I am satisfied with the revision and am happy to see this published. I am not sure whether this article could help to persuade criticized proponents but it would surely contribute to a more pluralistic stance in the literature.